# Efficacy and safety of short-term therapy with indigo naturalis for ulcerative colitis: An investigator-initiated multicenter double-blind clinical trial

Kan Uchiyama[1☯], Shinichiro Takami[1☯], Hideo Suzuki[2]*, Kiyotaka Umeki[3], Satoshi Mochizuki[4,5], Nobushige Kakinoki[6], Junichi Iwamoto[7], Yoko Hoshino[8], Jun Omori[9], Shunji Fujimori[9], Akinori Yanaka[10], Yuji Mizokami[2], Toshifumi Ohkusa[1,11]

1 Division of Gastroenterology and Hepatology, Department of Internal Medicine, The Jikei University Kashiwa Hospital, Kashiwa, Chiba, Japan, 2 Department of Gastroenterology, Faculty of Medicine, University of Tsukuba, Tsukuba, Ibaraki, Japan, 3 Department of Gastroenterology, Chiba-Nishi General Hospital, Matsudo, Chiba, Japan, 4 Division of Gastroenterology, Department of Internal Medicine, Tokatsu Tsujinaka Hospital, Abiko, Chiba, Japan, 5 The Shinagawa Gut Clinic, Minato-ku, Tokyo, Japan, 6 Department of Gastroenterology, Hitachi General Hospital, Hitachi, Ibaraki, Japan, 7 Division of Gastroenterology, Department of Internal Medicine, Tokyo Medical University Ibaraki Medical Center, Inashiki-gun, Ibaraki, Japan, 8 Division of Gastroenterology, Department of Internal Medicine, Yatsu Hoken Hospital, Narashino, Chiba, Japan, 9 Division of Gastroenterology, Department of Internal Medicine, Nippon Medical School Chiba Hokusoh Hospital, Inzai, Chiba, Japan, 10 Department of Gastroenterology, Hitachi Medical Education and Research Center, University of Tsukuba Hospital, Hitachi, Ibaraki, Japan, 11 Department of Microbiota Research, Juntendo University Graduate School of Medicine, Bunkyo-ku, Tokyo, Japan

☯ These authors contributed equally to this work.
* hideoszk@md.tsukuba.ac.jp

**Data Availability Statement:** All the data necessary to replicate the study's findings are available within the paper. Further personal

## Abstract

### Introduction

Indigo naturalis (IN) is a blue pigment extracted from Assam indigo and other plants and has been confirmed to be highly effective for ulcerative colitis (UC) treatment in several clinical studies.

### Objective

We conducted a multicenter double-blind study to confirm the efficacy and safety of short-term IN administration.

### Methods

A multicenter, randomized controlled trial was conducted between December 2015 and October 2018 in our facilities. Forty-six patients with mild to moderate active UC (Lichtiger index: 5–10) were randomly assigned to the IN group or the placebo group and received 5 capsules (500 mg) twice a day for 2 weeks. We investigated the efficacy according to blood tests and the Lichtiger index before and after administration, and we also examined adverse events.

information cannot be provided in accordance with domestic arrangements.

**Funding:** The authors received no specific funding for this work.

**Competing interests:** The authors have declared that no competing interests exist.

## Results

The analysis included 42 patients (20 males, 22 females) with an average age of 45 years. Nineteen patients were assigned to the placebo group, and 23 were assigned to the IN group. After treatment administration, in the placebo group, no change in the Lichtiger index was observed (7.47 to 6.95, p = 0.359), and hemoglobin was significantly reduced (12.7 to 12.4, p = 0.031), while in the IN group, the Lichtiger index (9.04 to 4.48, p = 0.001) and albumin (4.0 to 4.12, p = 0.022) improved significantly. Mild headaches were observed in 5 patients and 1 patient in the IN and placebo groups, respectively.

## Conclusions

Short-term administration of IN is highly effective without serious adverse events such as pulmonary hypertension or intussusception and may prevent the occurrence of serious adverse events.

## Introduction

Ulcerative colitis (UC) is an intractable inflammatory bowel disease (IBD) whose incidence is rapidly increasing worldwide, including in Japan. Currently, Japan has the second largest number of UC patients in the world after the United States. Conventionally, drugs such as 5-aminosalicylic acid (5-ASA) preparations, steroids, and immunomodulators (azathioprine and 6-methylmercaptopurine) are used for IBD treatment. In recent years, biologics such as infliximab, adalimumab, ustekinumab, and vedolizumab; low molecular weight compounds such as tofacitinib; and calcineurin inhibitors such as cyclosporine and tacrolimus have emerged, and treatment options are diverse. However, treatments for intractable patients who are resistant to these remission induction treatments or for intolerant patients, such as those with allergies, have not been established. In such cases, some natural products are used as an alternative or optional treatment, but their actual status and effects are unknown.

Indigo naturalis (IN) is a blue pigment found in leaves and stems of plants such as Assam indigo (*Strobilanthes cusia* of the Acanthaceae family), false indigo (*Indigofera bungeana* Walp of the Fabaceae family), and woad (*Isatis tinctoria* of the Brassicaceae family). Chinese herbal medicines containing IN have been used mainly as antifever and anti-inflammatory drugs, e.g., in the form of enema treatments for UC patients [1]. In Japan, IN powder has been sold as a health food, and its efficacy has been widely known in some UC patients. Then, some case reports that revealed the efficacy of IN for UC were published [2–4]. Additionally, the Chinese herbal medicine Xileisan enema, which is mainly composed of IN, is used to treat ulcerative enteritis in China, and the effectiveness of suppositories has been reported in Japan [5]. In recent years, it has been reported that 8 weeks of oral administration of IN has been effective in patients with mild to moderate active UC [6], however serious adverse events including pulmonary arterial hypertension (PAH) [7] or intussusception [8] were reported in patients treated with IN for long periods. In this study, we evaluated the efficacy and safety of short-term remission induction therapy with IN in a multicenter, randomized controlled trial to draw conclusions with higher-quality scientific evidence. In addition, because we did not limit the voluntary purchase and continuation of IN, we also conducted an evaluation of subsequent outcomes after the 2 weeks observation period.

## Materials and methods

### Patient selection

The subjects were UC patients with mild to moderate UC activity who were intolerant or refractory to existing treatments and were undergoing outpatient treatment from December 2015 to October 2018. The first patient was enrolled in December 14th, 2015. The number of samples was calculated by assuming that the efficacy of IN was 80% and that of placebo was 40%, with $\alpha = 0.05$, $\beta = 0.2$ and power = 0.8, resulting in 22 cases in each group. The end date was determined by the collected sample size because approximately 20 patients in each group seemed sufficient according to another randomized trial [6] and further recruitment of subjects was considered difficult with the establishment and enforcement of the Clinical Research Act. For minors over the age of 16 and younger than 20 years of age, consent was obtained from the participants and their guardians. In particular, pediatric cases (16 years old and younger) and minor patients aged 16 to under 20 years of age who were not able to obtain consent from their parents or guardians were excluded.

The selection criteria were as follows: 16 years and older, performance status (Eastern Cooperative Oncology Group (ECOG)) of 0 or 1, active mild to moderate UC (Lichtiger index [9]: 5–10), outpatient visit, tolerant or refractory to existing treatments, hemoglobin (Hb)≧9 g/dl, aspartate aminotransferase (AST)≦facility standard, alanine aminotransferase (ALT)≦facility standard and serum creatinine≦facility standard.

The exclusion criteria were as follows: patients who had undergone any UC intervention within the past 2 weeks, including adding or increasing 5-ASA and steroids or using topical products, or patients who had received or increased remission induction therapy, such as thiopurines, infliximab, adalimumab, and calcineurin inhibitors, within the past three months. In addition, pregnant women, women who might be pregnant, and patients who had ever undergone natural product treatment, including treatment with IN, were excluded. Data were collected at each facility.

### Study design

Case registration was performed using the clinical research support system HOPE eACReSS (FUJITSU, Tokyo, Japan). This system was used to perform minimization randomization into the IN group or placebo group using gender and the Lichtiger index before administration. Using this system, the score of the Lichtiger index before and after administration, various blood test data including WBCs, Hb, Alb, CRP, ESR, and presence or absence of adverse events including physical problems and biochemical test were compared.

The IN powder (JAN code 4987359922404) used in this study was purchased from Uchidawakanyaku, Ltd. (Tokyo, Japan). Quantified compounds in the IN are described in S1 Table in reference 6. A total of 100 mg of IN or placebo, rice starch, was put in each capsule. These were formulated together with a dehumidifier in a plastic bottle and delivered to each facility. The IN and placebo were kept in sealed containers, avoiding high temperatures and humidity, and those whose expiration date had passed were not used. Subjects took 5 capsules (500 mg) of IN or placebo once a day for 2 weeks in addition to their usual medications.

Blood tests were performed on the day before and 2 weeks after administration, and complete blood counts, total protein, albumin, AST, ALT, creatinine and C-reactive protein (CRP) were required as test items. Due to the short administration period of 2 weeks, endoscopic observation before and after was not mandatory. Subjective symptoms such as headache or shortness of breath associated with PAH were carefully monitored by questioning.

As the primary endpoint, the disease activity index (Lichtiger index) at 2 weeks after administration was defined as follows: a reduction of more than 30% compared to baseline was defined as a "response" and a reduction of more than 50% was defined as a "marked response". As a secondary endpoint, we evaluated the specifics and prevalence rates of newly observed physical adverse events and blood test values during the administration period. In some patients who voluntary purchase and continuation of IN after the 2 weeks, the outcome and Lichtiger index at 24 weeks after the start of IN administration was investigated. The adverse events were also monitored and brain natriuretic peptide (BNP) was measured regularly in order to detect PAH early.

The study protocol is available as supplementary information.

## Statistical analyses

For statistical analysis, we employed SPSS for Windows software (SPSS Japan, Tokyo, Japan). The effect of treatment on response rate was assessed using the chi-square test. The differences in Lichtiger index between groups was assessed using the unpaired two-sided t-test, and the difference within groups over time was assessed using the paired two-sided t-test. Statistical significance was declared at the 0.05 level.

## Ethics

This study was approved by the Jikei University Hospital Ethics Committee (26–364) in June 8th, 2015 and registered in the University Hospital Medical Information Network Clinical Trial Registry (UMIN000019103, available at http://www.umin.ac.jp/ctr/) which is one of Japan Primary Registries Network (JPRN) approved by WHO. Written informed consent was obtained from all participants.

## Results

Forty-seven patients were treated for UC between December 2015 and October 2018 and enrolled in this study. One patient was excluded due to exclusion criteria violation. Forty-six patients were randomly assigned to the placebo group or IN group. One case of low compliance, one case of consent withdrawal, and two cases of disease exacerbation were excluded after randomization. Finally, 19 patients were assigned to the placebo group, and 23 were assigned to the IN group (Fig 1).

Of the 42 patients, 20 patients were males, and 22 patients were females. The mean age was 45.9±16.5 (18 to 79) years, and the distribution of disease types was as follows: 26 cases of pancolitis, 13 cases of left-sided colitis, and 3 cases of proctitis. The baseline characteristics of each group are shown in Table 1. At baseline, there were no significant differences between the placebo group and the IN group except for in the Lichtiger index. Before administration, the mean Lichtiger index in the IN group was significantly higher than that in the placebo group (9.04±1.92 vs. 7.47±1.43, p = 0.0053).

After treatment administration, in the placebo group, no change in the Lichtiger index was observed (7.47±1.43 to 6.95±2.61, p = 0.359), and Hb was significantly reduced (12.7±1.85 g/dl to 12.4±1.59 g/dl, p = 0.031), while in the IN group, albumin (4.0±0.46 g/dl to 4.12±0.50 g/dl, p = 0.022) and the Lichtiger index (9.04±1.92 to 4.48±2.21, p = 0.001) improved significantly (Fig 2, Table 2).

Regarding comparisons between the two groups, the mean Lichtiger index in the IN group was significantly higher than that in the placebo group before administration, although after 2 weeks of administration, the mean Lichtiger index in the IN group was significantly lower than that in the placebo group (p = 0.0019) (Fig 3).

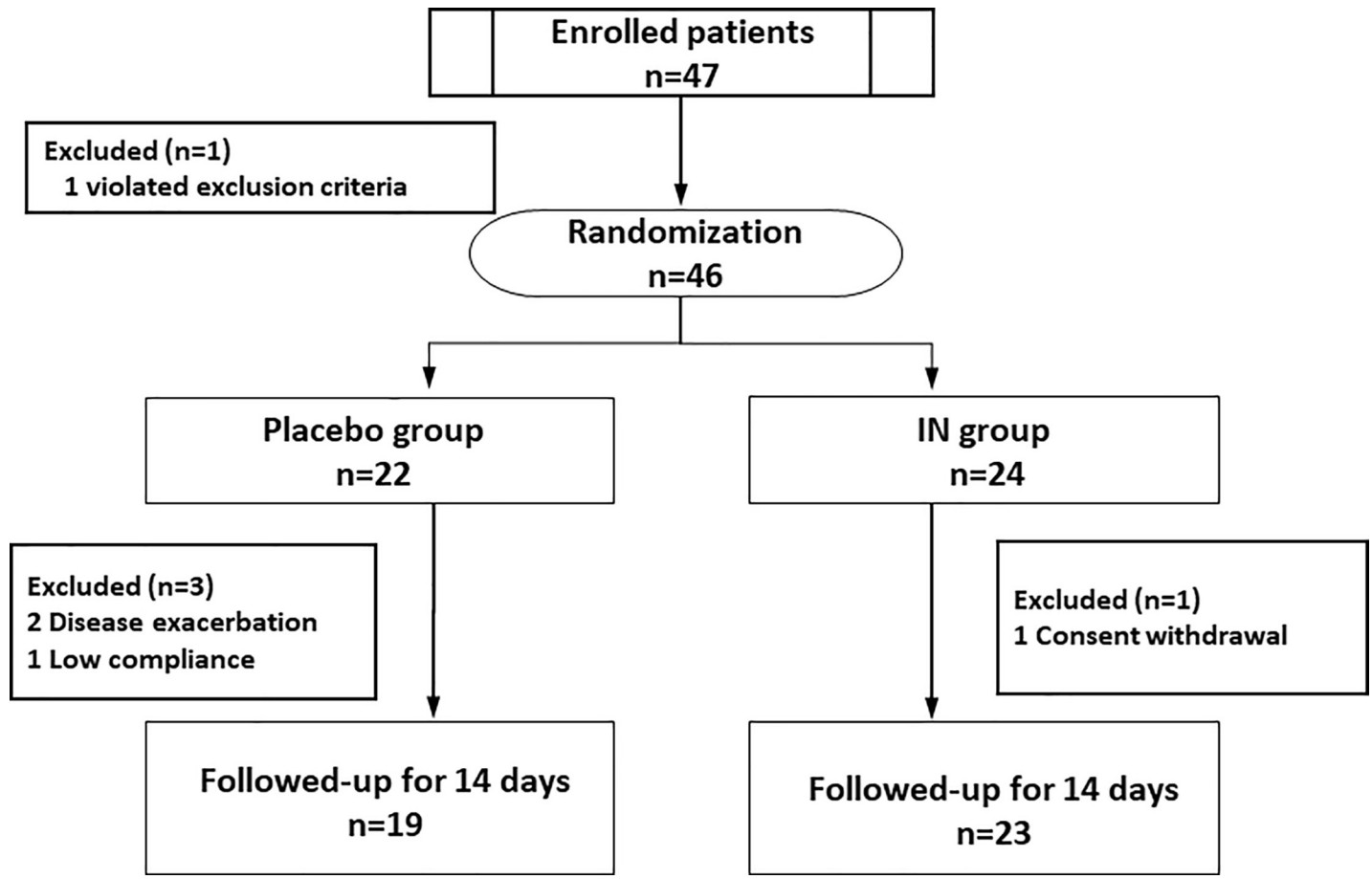

**Fig 1. The research flow diagram.** IN: indigo naturalis.

The response rates of each group were 26.3% (5 of 19 patients) in the placebo group and 82.6% (19 of 23 patients) in the IN group, with a significant difference (p = 0.0003). The marked-response rate in the IN group was significantly higher than that in the placebo group (60.9%; 14 of 23 vs. 5.3%; 1 of 19, p = 0.0002) (Fig 4).

The comparisons of baseline characteristics between responders and nonresponders could not be analyzed because only 3 of 23 patients were nonresponders in the IN group. Therefore, we analyzed the differences between marked responders (n = 14) and nonmarked responders (n = 9). No significant differences were found (S1 Table).

Regarding adverse events, mild headaches were observed in 5 of 23 patients (21.7%) in the IN group; however, there was no statistically significant difference from the respective results in the placebo group (1 of 19 patients, 5.26%; p = 0.141). Regarding biochemical test data, although AST, ALT, and gamma glutamyl transpeptidase (GTP) in the IN group were significantly elevated (17.0±5.60 IU/L to 20.0±5.16 IU/L; p = 0.015, 13.0±7.70 IU/L to 18.1±8.60 IU/L; p = 0.004, 18.3±9.96 IU/L to 21.5±10.1 IU/L; p = 0.004, respectively), all were within normal ranges (Table 2). No serious adverse events, such as PAH or intussusception, were observed after short-term administration for 2 weeks.

Most participants wished to continue taking their self-purchased IN after 2 weeks of administration in this study. Indeed, 32 of 42 (76.2%) participants actually purchased IN and continued taking it. When 24-week remission was defined as a Lichtiger index ≤3, 30 of 32 patients

**Table 1. Baseline characteristics.**

| | | Placebo (n = 19) | IN (n = 23) | p-value |
|---|---|---|---|---|
| Age | Mean±SD | 50.1±13.8 | 42.5±18.0 | 0.14† |
| Sex | Male (%) | 9 (47.3) | 11 (47.8) | 0.61‡ |
| Disease type | | | | 0.72‡ |
| pancolitis | | 11 (57.9) | 15 (65.2) | |
| left-sided colitis | n (%) | 7 (36.8) | 6 (26.1) | |
| proctitis | | 1 (5.3) | 2 (8.7) | |
| Duration (months) | Mean±SD | 114.7±90.8 | 76.3±67.8 | 0.12† |
| Prior treatment | | | | |
| 5-ASA | n (%) | 17 (89.5) | 20 (87.0) | 0.23‡ |
| PSL | n (%) | 2 (10.5) | 6 (26.1) | 0.19‡ |
| AZA | n (%) | 4 (21.1) | 3 (13.4) | 0.39‡ |
| BIO | n (%) | 0 (0) | 2 (8.7) | 0.29‡ |
| Lichtiger index | Mean±SD | 7.47±1.429 | 9.04±1.918 | 0.0053† |
| WBCs | Mean±SD | 5579±2646.6 | 5860±2880.5 | 0.746† |
| CRP | Mean±SD | 0.42±0.570 | 0.38±0.742 | 0.837† |
| ESR | Mean±SD | 21.5±18.33 | 14.0±11.91 | 0.138† |
| Hb | Mean±SD | 12.7±1.85 | 13.4±1.46 | 0.214† |
| Alb | Mean±SD | 3.9±0.34 | 4.0±0.46 | 0.509† |

† t-test

‡ Chi-square test.

5-ASA: 5-aminosalicylic acid; PSL: prednisolone; AZA: azathioprine; BIO: biologics; WBCs: white blood cells; CRP: C-reactive protein; ESR: erythrocyte sedimentation rate; Hb: hemoglobin; Alb: albumin.

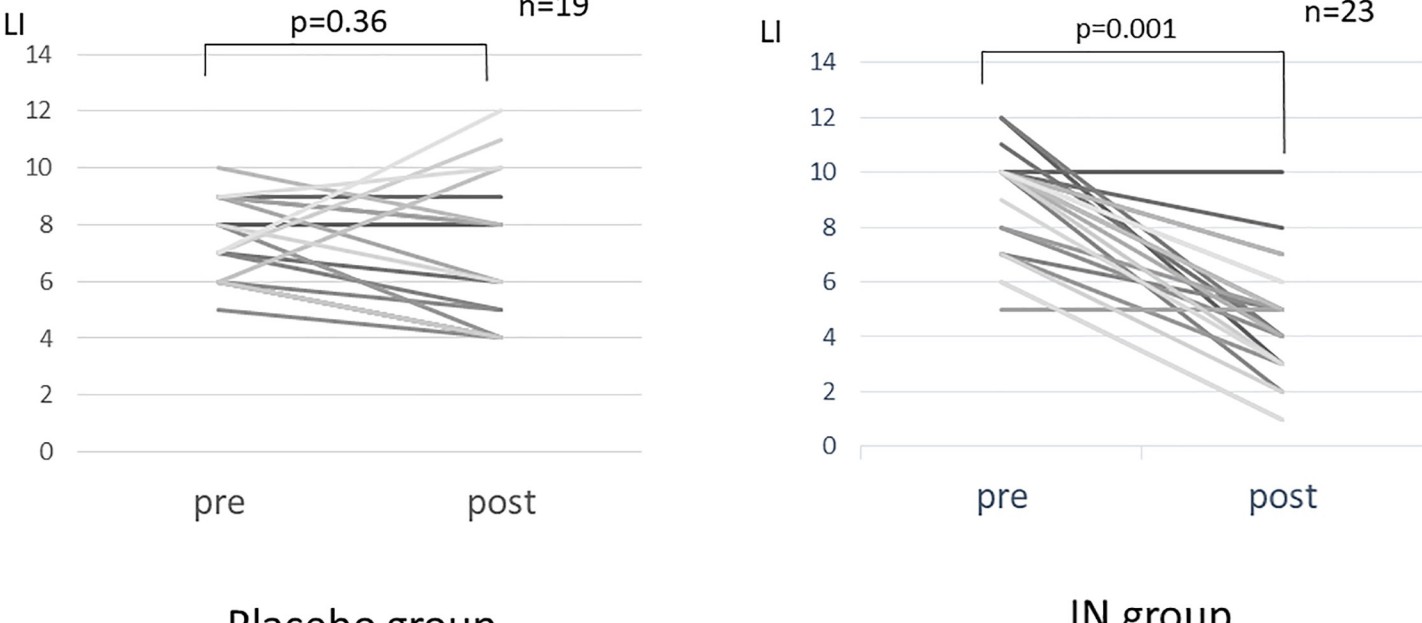

**Fig 2. Individual changes in the Lichtiger index before and after the intervention.** IN: indigo naturalis; LI: Lichtiger index.

**Table 2. Data comparison before and after the intervention in each group.**

| | Placebo group | | | IN group | | |
|---|---|---|---|---|---|---|
| | pre | post | p | pre | post | p |
| WBCs | 5679±2647 | 5445±3385 | 0.828 | 5860±2880 | 5866±2607 | 0.852 |
| Hb | 12.7±1.85 | 12.4±1.59 | 0.031* | 13.4±1.46 | 13.4±1.42 | 0.905 |
| Plt | 32.1±7.45 | 42.2±40.4 | 0.319 | 31.4±10.6 | 32.1±10.9 | 0.475 |
| AST | 19.4±5.98 | 17.1±4.93 | 0.053 | 17.0±5.60 | 20.0±5.16 | 0.015* |
| ALT | 15.0±6.20 | 13.1±4.08 | 0.056 | 13.0±7.70 | 18.1±8.60 | 0.004** |
| γGTP | 22.4±22.6 | 21.6±21.5 | 0.175 | 18.3±9.96 | 21.5±10.1 | 0.004** |
| TP | 7.15±0.55 | 7.04±0.65 | 0.17 | 7.03±0.46 | 7.22±0.52 | 0.055 |
| Alb | 3.91±0.34 | 3.87±0.35 | 0.26 | 4.00±0.46 | 4.12±0.50 | 0.022* |
| BUN | 11.9±4.65 | 12.1±3.23 | 0.733 | 9.95±2.88 | 11.1±2.93 | 0.06 |
| Cr | 0.70±0.16 | 0.68±0.15 | 0.108 | 0.70±0.17 | 0.67±0.15 | 0.284 |
| ESR | 21.5±18.3 | 20.9±18.0 | 0.305 | 14.1±11.9 | 15.4±16.6 | 0.575 |
| CRP | 0.42±0.57 | 0.26±0.23 | 0.107 | 0.38±0.74 | 0.50±1.65 | 0.596 |
| Lichtiger index | 7.47±1.43 | 6.95±2.61 | 0.359 | 9.04±1.92 | 4.48±2.21 | 0.001*** |

Paired t-test

*p<0.05

**p<0.01

***p<0.001.

WBCs: white blood cells; Hb: hemoglobin; Plt: platelets; AST: aspartate aminotransferase; ALT: alanine aminotransferase; γGTP: gamma glutamyl transpeptidase; TP: total protein; Alb: albumin; UN: urea nitrogen; Cr: creatinine; ESR: erythrocyte sedimentation rate; CRP: C-reactive protein.

were in remission, and the status of the remaining 2 patients was unknown due to hospital interruption (Fig 5). At 24 weeks, there were no severe adverse events, such as PAH or intussusception, and headache (3 cases), constipation (1 case), and palpitations (1 case) were the only adverse events reported. All adverse events improved or disappeared due to IN discontinuation or dose reduction.

## Discussion/Conclusion

Although there have been some reports on the efficacy of Chinese herbal medicines, including IN and Xileisan, for active UC [2–5], the mechanisms of their therapeutic effects remain unknown. It has been reported that the effect on Crohn's disease was not as high as on UC [10]. We reported the hydroxyl radical scavenging effect of IN in our previous case report [3], and other previous review reports showed that the indigo and indirubin included in IN act as aryl hydrocarbon receptor (AhR) ligands and can potentially promote mucosal healing by stimulating mucosal type 3 innate lymphoid cells to produce interleukin-22 [1, 11]. Furthermore, another study revealed that IN treatment significantly decreased the infiltration of macrophages and the production of inflammatory cytokines, such as TNFα, IL-1β, and IL-6, in the colon tissue of dextran sulfate sodium-treated mice [12]. On the other hand, severe adverse events such as PAH [7] and intussusception due to colonic wall thickening [8] were reported sporadically. Furthermore, colitis due to IN has also been reported [13–15]. However, these reports were from single-center observational studies or case reports, and the exact effectiveness or incidence of adverse events related to IN treatment has remained unclear for a long time. Prior to our present report, Naganuma et al. reported the efficacy of 8 weeks of IN administration for UC in a multicenter randomized controlled trial [6]. In this report, participants were assigned randomly into three IN groups (0.5 g/day, 1.0 g/day, or 2.0 g/day) and a

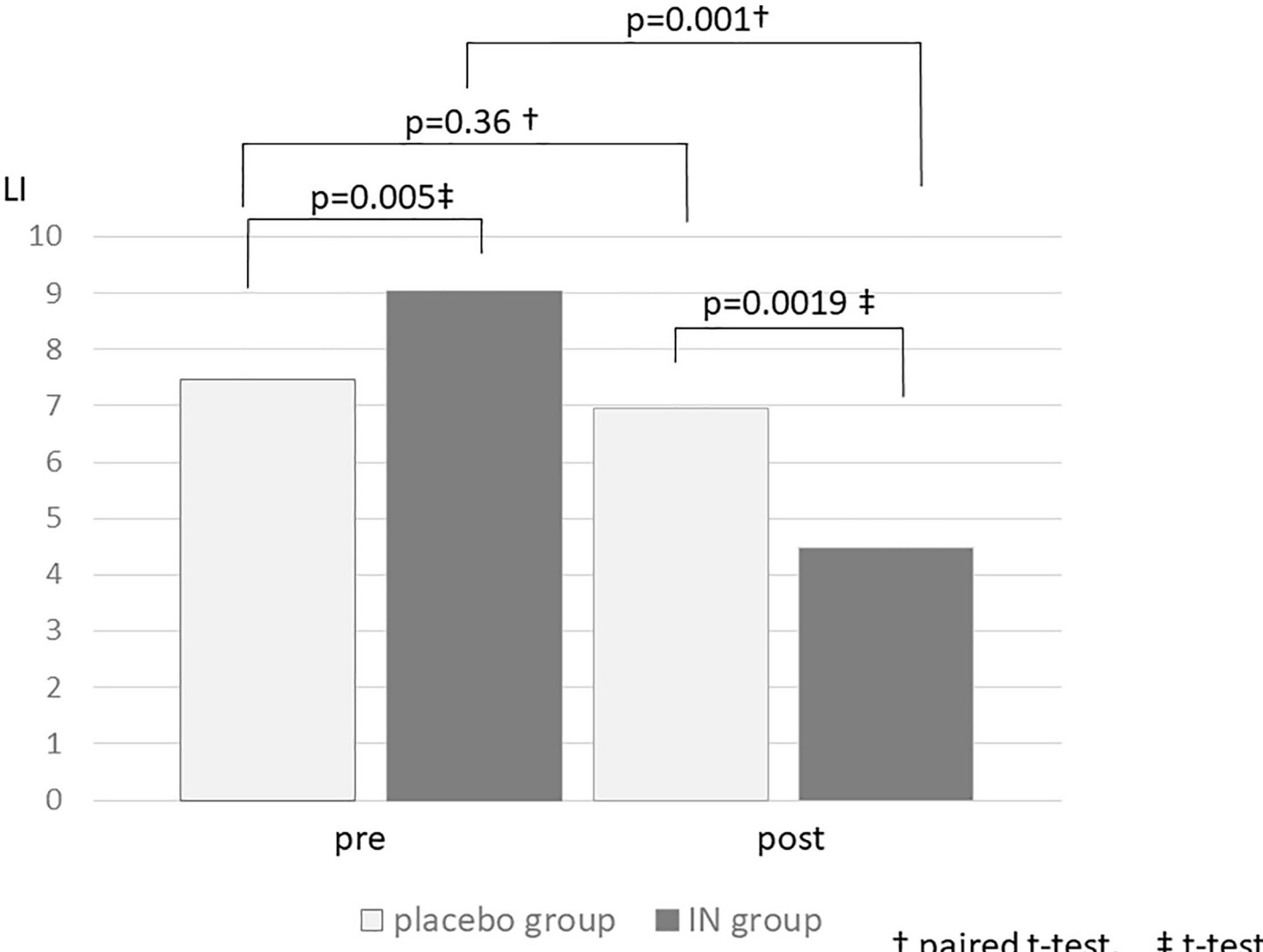

**Fig 3. The average change in the Lichtiger index before and after the intervention.**

placebo group. Although this trial was terminated due to an external reason, a report of a PAH in a patient who used self-purchased IN for 6 months, the study revealed a significant, dose-dependent response in the proportions of patients with clinical responses (13.6% with a clinical response to placebo; 69.6% to 0.5 g IN; 75.0% to 1.0 g IN; and 81.0% to 2.0 g IN, respectively). Furthermore, in a Japanese nationwide questionnaire-based survey, the incidence of adverse events due to IN was reported [16]. According to this report, the rate of patients who used IN was 877 (1.8%) of 49,320 patients with UC, and adverse events were reported in 91 patients (10.4%). Most of those adverse events were mild, such as liver dysfunction (n = 40; 4.6%), gastrointestinal symptoms (n = 21; 2.4%), and headache (n = 13; 1.5%); serious adverse events such as PAH and intussusception were observed in 11 patients (1.25%) and 10 patients (1.14%), respectively. Most cases of PAH were observed in patients treated with IN for more than 24 weeks, but intussusception occurred 3 to 8 weeks after the initiation of IN, and intussusception tended to be shorter. Regarding the dose of IN, it was reported that intussusception occurred at a dose of 0.5–2.0 g/day, and the doses for PAH cases were not specified. As

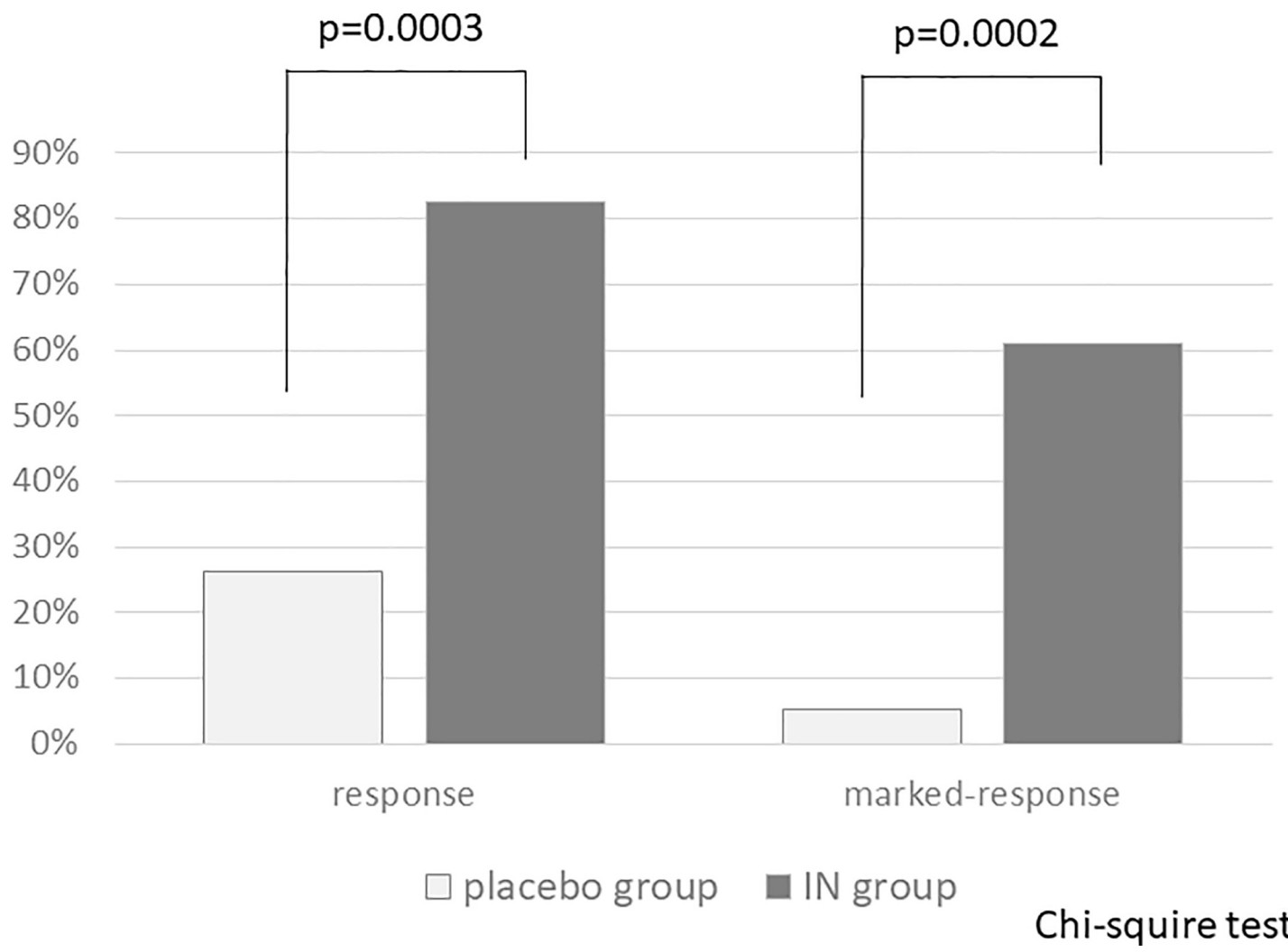

**Fig 4. Comparisons of response and marked response rates.** IN: indigo naturalis.

described above, the mechanism of occurrence of such serious adverse events is unknown, but it is possible that indigo and indirubin contained in IN act as AhR ligands. Previously, AhR knockout mice were reported to have abnormal vascular development in the liver [17]. There is a report that AhR induces vascular endothelial growth factor (VEGF) expression [18], and it is possible that the exogenous ligand IN may have some effect on angiogenesis in lung tissue and intestinal mucosa. It is not clear whether these adverse events occur in a dose-dependent manner, but in the case of severe mucosal damage, IN may be more absorbed even at low doses. Therefore, our results are meaningful to considered safer to administer IN for as short of a duration as possible to avoid these serious adverse events, and the same strategy to induce remission in short term is similar to that of prednisolone and tacrolimus.

The efficacy and safety of long-term administration of IN is a matter of debate. At the time our study was planned, although the incidence of these adverse events was known, their frequency was unknown. As a result, in this study, though the administration period was as short as 2 weeks, some patients continue to take IN voluntarily after 2 weeks of observation periods. The outcome of 24weeks of administration was sufficient. IN seems to be expected not only to

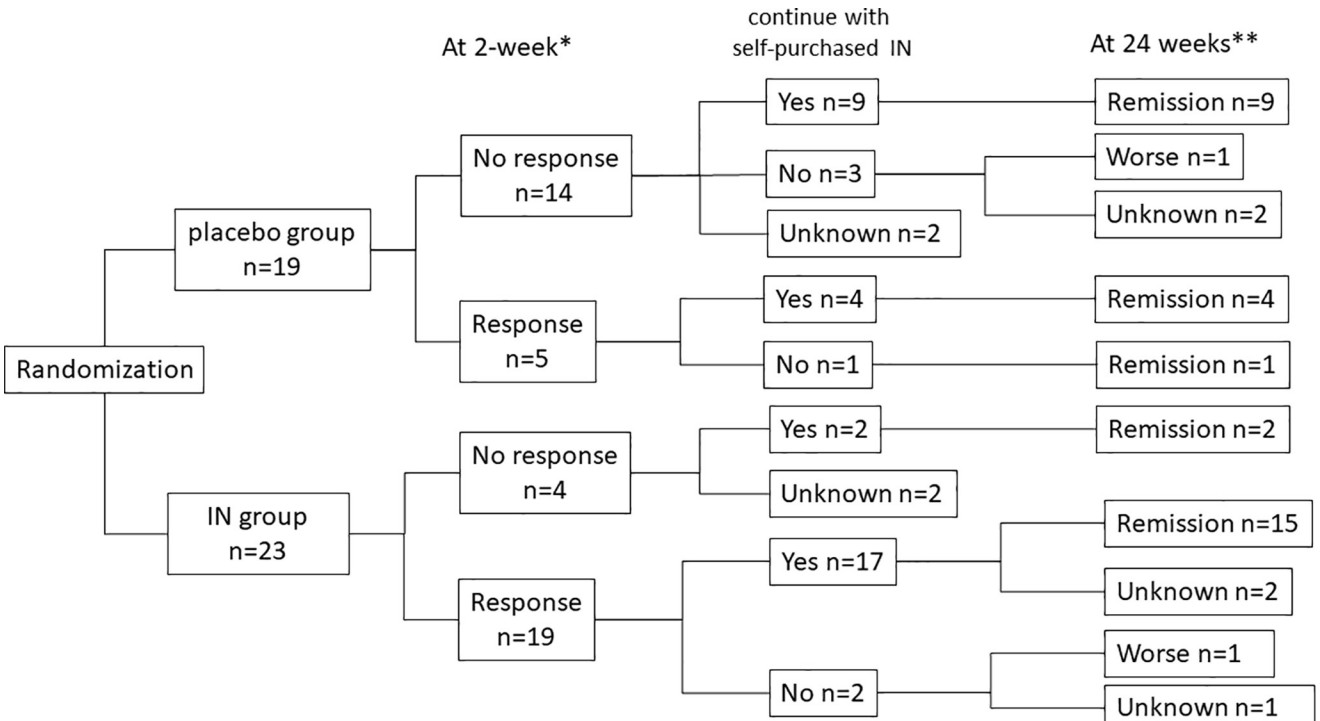

**Fig 5. Survey results 24 weeks after the start of the research.** IN: indigo naturalis; LI: Lichtiger index.

induce remission but also to maintain remission; however, another rigorous double-blind study is needed to confirm long-term efficacy and safety.

At present, IN should be used only in refractory cases where it is difficult to induce or maintain remission with other agents, such as in cases of 5-ASA allergies. When using IN for treatment, the attending physician should always be aware of the occurrence of adverse events and should provide sufficient information to patients using IN. It is necessary to clarify the relationship between the administration period, efficacy, and adverse events through more extensive research as well as clinical experience and establish appropriate monitoring methods.

The limitations of this study are that the small number of patients and the short duration of administration could not indicate long-term efficacy or safety. Furthermore, one of the limitations is that endoscopic evaluation could not be performed in a short period of time. Despite the small number of patients, it is worth noting that a significant difference was confirmed after administration for only 2 weeks.

In conclusion, IN was highly effective even after short-term administration for 2 weeks, and no serious adverse events were observed. Although short-term administration might be able to avoid serious adverse events, such as PAH or intussusception, it is necessary to develop an appropriate method for safe use in the future.

## Supporting information

**S1 Table. Comparison between marked responders and nonmarked responders.**
(DOCX)

**S1 File.**
(DOCX)

**S2 File.**
(DOCX)

## Acknowledgments

We would like to express our gratitude to Visiting Professor Shigeru Kageyama, the former director of the Clinical Research Support Center of Jikei University, for his guidance and efforts in obtaining approval for the rigorous IRB examination for realizing this research. We would also like to thank the following physicians for their generous support during this research: Toshihide Ohmori, Ohmori Toshihide Gastrointestinal Clinic; Junji Kasanuki, Funabashi Minato Clinic; Yoshinori Hiroshima, Hitachinaka General Hospital; Mariko Wakayama, Koyama Memorial Hospital; Ryuzo Murai, Onaka Clinic; Takashi Mamiya, Ryugasaki Saiseikai Hospital; Hiroyasu Ishida, National Hospital Organization Mito Medical Center; Hiroshi Kashimura, Mito Saiseikai General Hospital; Izumi Shirato, Tokyo Women's Medical University Yachiyo Medical Center; and Mitsuaki Hirose, National Hospital Organization Kasumigaura Medical Center.

## Author Contributions

**Conceptualization:** Kan Uchiyama, Hideo Suzuki, Yuji Mizokami, Toshifumi Ohkusa.

**Data curation:** Kan Uchiyama, Hideo Suzuki.

**Formal analysis:** Shinichiro Takami, Hideo Suzuki.

**Investigation:** Kan Uchiyama, Hideo Suzuki.

**Methodology:** Kan Uchiyama.

**Project administration:** Kan Uchiyama, Shinichiro Takami, Hideo Suzuki, Kiyotaka Umeki, Satoshi Mochizuki, Nobushige Kakinoki, Junichi Iwamoto, Yoko Hoshino, Jun Omori, Shunji Fujimori.

**Supervision:** Akinori Yanaka, Yuji Mizokami, Toshifumi Ohkusa.

**Writing – original draft:** Kan Uchiyama, Shinichiro Takami.

**Writing – review & editing:** Hideo Suzuki, Kiyotaka Umeki, Satoshi Mochizuki, Nobushige Kakinoki, Junichi Iwamoto, Yoko Hoshino, Jun Omori, Shunji Fujimori, Akinori Yanaka, Yuji Mizokami, Toshifumi Ohkusa.

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
