## [Decision Letter · Decision Letter 0]

1 Sep 2020

PONE-D-20-17829

Efficacy and safety of short-term therapy with indigo naturalis for ulcerative colitis: An investigator-initiated multicenter double-blind clinical trial

PLOS ONE

Dear Dr. Suzuki,

Thank you for submitting your manuscript to PLOS ONE. After careful consideration, we feel that it has merit but does not fully meet PLOS ONE’s publication criteria as it currently stands. Therefore, we invite you to submit a revised version of the manuscript that addresses the points raised during the review process.

Two experts in the field reviewed your manuscript. One reviewer had concerns about conflicting data in the field. The other reviewer had several follow-up questions. Please address all comments given by both reviewers, revise your manuscript appropriately, and resubmit.

We look forward to receiving your revised manuscript.

Kind regards,

Susan Hepp

Academic Editor

PLOS ONE

Journal Requirements:

2. Please include a sample size and power calculation in your methods section

Reviewers' comments:

Reviewer's Responses to Questions

**Comments to the Author**

1. Is the manuscript technically sound, and do the data support the conclusions?

Reviewer #1: Yes

Reviewer #2: Yes

2. Has the statistical analysis been performed appropriately and rigorously? 

Reviewer #1: Yes

Reviewer #2: Yes

3. Have the authors made all data underlying the findings in their manuscript fully available?

Reviewer #1: Yes

Reviewer #2: Yes

4. Is the manuscript presented in an intelligible fashion and written in standard English?

Reviewer #1: Yes

Reviewer #2: Yes

5. Review Comments to the Author

Reviewer #1: Our recommendation on this manuscript is to refuse

At present, international and mainland China clinical trials show that the effect of chemical monomer extracts of traditional Chinese medicine on ulcerative colitis is not satisfactory, indicating that the prospects of this kind of drug treatment for colitis are not clear, including the classical curcumin, baicalin and so on. Therefore, the above drugs have not been marketed in mainland China. This objective phenomenon for many years proves that these drugs do have defects. In view of the following deficiencies in this article, it is recommended that the manuscript be rejected

1) Lack of repetitive testing.

2) The sample size of patients is small.

3) The administration time is short, which can not explain the long-term efficacy or safety.

4) Endoscopic evaluation cannot be performed in a short period of time.

However, the experimental content in this manuscript also has some advantages, as follows：

1) Using Lichtiger index as the judgement of curative effect is more consistent with the results of the trail.

2) They analyzed the differences between marked responders and nonmarked responders since only 3 of 23 patients were nonresponders in the IN group

3) Before this trial, they had studied and reported the mechanisms of the IN therapeutic effects in their previous case report, although the exact mechanisms remain unknown.

4) This trial deeply explored the adverse events, not only summarized all the adverse events that would occur, but also discussed the mechanism of occurrence of serious adverse event, particularly the relationship between the dose of IN and the occurrence of severe adverse events, and considered safer to administer IN for as short of a duration as possible to avoid these serious adverse events.

Reviewer #2: In this randomizes study, authors showed the excellent efficacy of IN treatment at 2 weeks. They showed the remarkable improvement of Lichtiger index, hemoglobin, and albumin after IN treatment compared to placebo group. Although IN is not commonly used drug and there is safety concern, there was no serious adverse event even at 24 weeks. Although several studies already showed these results, IN treatment for UC is not widespread. Gastroenterologist will be interest in this study.

Minor

Introduction

There has been already published several academic researches about IN in the treatment for UC. The traditional use of IN should be short (or may not be needed).

Result

1.They showed baseline characteristics and said there were no significant difference between two group. How about endoscopic score?

2.They described prior treatment. I am interested to see if they could stop these treatments after IN treatment.

3.Pulmonary arterial hypertension is serious adverse event. So, they should pay attention for PAH. Did they monitor something in the patients to find adverse event early.

Discussion

1.Given the mechanism of IN they described, is IN useful for other IBD?

2.I understand main outcome of this study is 2 weeks data, however, usually data (even at 24 week) should described in result section. Reader will be interest in the results at 24 weeks.

The following are checklist.

1) Outcomes (Item 6a)

Outcomes are clear.

2) Sample size (Item 7a)

Although sample size is small, they determine sample size appropriately.

3) Sequence generation (Item 8a)

Method used to generate random allocation sequence

4) Allocation concealment (Item 9)

Case registration was performed using appropriate algorithm.

5) Blinding (Item 11a)

Double blind.

6) Outcomes and estimation (Item 17a/b)

Well described.

7) Harms (Items 19)

Adverse events were described.

8) Registration (Item 23)

UMIN number was described.

9) Protocol (Item 24)

Web site is available.

10) Funding (Item 25)

Sources of funding and other support (such as supply of drugs) and role of funders

Are (1) the funding sources, and (2) the role of the funder(s) described?

I could not find the sentences about funding. If there was funding, please described it.

6. PLOS authors have the option to publish the peer review history of their article (what does this mean?). If published, this will include your full peer review and any attached files.

Reviewer #1: No

Reviewer #2: No

---

## [Author Response · Author response to Decision Letter 0]

11 Sep 2020

Please find the file named "Response to Reviewers".

---

## [Decision Letter · Decision Letter 1]

30 Sep 2020

PONE-D-20-17829R1

Efficacy and safety of short-term therapy with indigo naturalis for ulcerative colitis: An investigator-initiated multicenter double-blind clinical trial

PLOS ONE

Dear Dr. Suzuki,

Thank you for submitting your manuscript to PLOS ONE. After careful consideration, we feel that it has merit but does not fully meet PLOS ONE’s publication criteria as it currently stands. Therefore, we invite you to submit a revised version of the manuscript that addresses the points raised during the review process.

We look forward to receiving your revised manuscript.

Kind regards,

Wan-Long Chuang, M.D., Ph.D.

Academic Editor

PLOS ONE

Reviewers' comments:

Reviewer's Responses to Questions

**Comments to the Author**

1. If the authors have adequately addressed your comments raised in a previous round of review and you feel that this manuscript is now acceptable for publication, you may indicate that here to bypass the “Comments to the Author” section, enter your conflict of interest statement in the “Confidential to Editor” section, and submit your "Accept" recommendation.

Reviewer #1: (No Response)

Reviewer #2: (No Response)

Reviewer #3: (No Response)

2. Is the manuscript technically sound, and do the data support the conclusions?

Reviewer #1: Yes

Reviewer #2: Yes

Reviewer #3: Yes

3. Has the statistical analysis been performed appropriately and rigorously? 

Reviewer #1: Yes

Reviewer #2: Yes

Reviewer #3: Yes

4. Have the authors made all data underlying the findings in their manuscript fully available?

Reviewer #1: Yes

Reviewer #2: Yes

Reviewer #3: Yes

5. Is the manuscript presented in an intelligible fashion and written in standard English?

Reviewer #1: Yes

Reviewer #2: Yes

Reviewer #3: Yes

6. Review Comments to the Author

Reviewer #1: We basically agree with the author’s reply. But there are still several issues that need to be put forward

1、The short duration of administration could not indicate long-term efficacy or safety. Lack of long-term patient with follow-up is unable to identify whether the patient has recurrence or not. For the conventional patients with UC, the short-term treatment with IN is of little significance. In contrast, more studies currently focus on the maintenance treatment with IN. In addition, many scholars have studied IN for the treatment-refractory patients with UC. In conclusion, research into short-term therapy with IN is less innovative

2、The IN clinical trials for the treatment of UC, relatively complete is Naganuma et al.published in Gastroenterology the Efficacy of Indigo naturalis in a Multicenter Randomized Controlled Trial of Patients with Ulcerative Colitis. What is the innovation of your trial compared with Naganuma et al. and other clinical trials.

3、However, IN should not yet be used because of the potential for adverse effects, including pulmonary arterial hypertension or intussusception, and this could be the main reason for that China has not yet used IN for UC patients in large-scale. And no serious adverse events were observed after short-term administration for 2 weeks as mentioned in the author’s manuscript. Whether this could be an advantage as a short-term treatment is worth exploring. It may be significant to study the optimal treatment course and dosage of IN for UC on the premise of minimizing the occurrence of severe adverse reactions.

Reviewer #2: The revised version is improved. Most parts are written well.

I apologize for your misunderstanding. I mentioned last time that the description in the introduction section is not academic. There have already been many academic studies about IN. The descriptions about "the plant", “kai Ban Ben Cao” and “Furikake” were not related to this study. Please shorten the description of traditional use of IN.

“the endoscopy was not required” added in the revise is not logical. The endoscopy was not required in the “study design” (especially at 2 weeks). This paper described information at entry, 2 weeks, and 24 weeks. Please write a suppressed expression instead of “the endoscopy was not required”.

Reviewer #3: In general, this is an interesting manuscript, well-executed and well-written.

*** The following are general comments or specific issues: ***

1) The results after 24 weeks are of interest. However, they are not presented properly in the paper. Since they are somewhat observational, it is important to clearly describe why and how they were obtained, as well as the results and their potential interpretation.

Currently, these pieces of information are all located in the Discussion section, which is incorrect. Please provide:

* A clear description of the rationale for additional assessment in the Background section (i.e., Lines 235-243),

* A clear description of methods of data collection and data actually collected in the Methods section (i.e., Lines 244-247)

* The results in the Results section (i.e., Supplementary Figure 1 and Lines 247-256), and

* Interpretation in the Discussion section (i.e., Lines 256-264).

* Supplementary Figure 1 should then be promoted to "Figure" status.

Again, the only issue here is that the provenance of the data should be clearly explained, the data should be presented clearly, and interpretation should be made in light of any potential biases.

2) In general, the statistical analysis is serviceable. One can argue endlessly about whether the authors ought to perform an analysis of covariance or a repeated measures analysis, but it seems very unlikely to change the key results. Additionally, the raw data for Lichtiger index are shown, so the reader may judge.

3) The discussion of sample size is moot at this point, since the study is complete and sample size is now determined. Naturally, it would be better to have more patients on study, but we must do what we can. The authors' response on this issue is satisfactory.

4) Although it gets the job done, the analysis presented in Lines 181-184 is somewhat crude. Table 3 should probably be demoted to "Supplementary Table" status.

The authors do not need to amend the analysis. However, they may wish to perform simple linear modelling of the Lichtiger index on these covariates instead of dichotomizing the data. This method should provide more power to detect associations. If not already done, the authors should definitely plot the Lichtiger index across each of these covariates to screen for patterns.

*** The following are minor suggestions: ***

* Line 68: Change to "or for intolerant patients".

* Line 110: In this case, does "intervention" refer to "any intervention" or "any UC intervention"?

* Lines 119-120: Change to "This system was used to perform minimization randomization using the Lichtiger index before and after administration, [SPECIFY BLOOD TEST DATA USED], and presence or absence of adverse events [SPECIFY SPECIFICS OF ADVERSE EVENTS THAT WERE USED]." Then, probably Lines 121-125 may be deleted.

* Line 145-146: Change to "The effect of treatment on response rate was assessed using the chi-square test. The differences in Lichtiger index between groups was assessed using the unpaired two-sided t-test, and the difference within groups over time was assessed using the paired two-sided t-test. Statistical significance was declared at the 0.05 level." Please check that t-tests were performed using the two-sided version.

* In Line 214, the authors describe a "dose-dependent linear trend". This seems a bit of a stretch, since the data given seem to describe a sharp rise and then a plateau instead. I suggest to amend this to a "dose-dependent response".

* Line 230: Change to "and it is possible that the exogenous".

* Please harmonize the use of colors, labels, and the placement of legends (bottom or right) among the figures.

7. PLOS authors have the option to publish the peer review history of their article (what does this mean?). If published, this will include your full peer review and any attached files.

Reviewer #1: No

Reviewer #2: No

Reviewer #3: No

---

## [Author Response · Author response to Decision Letter 1]

10 Oct 2020

Please find Responce to reviewer2 file.

---

## [Editor Report · Decision Letter 2]

14 Oct 2020

Efficacy and safety of short-term therapy with indigo naturalis for ulcerative colitis: An investigator-initiated multicenter double-blind clinical trial

PONE-D-20-17829R2

Dear Dr. Suzuki,

We’re pleased to inform you that your manuscript has been judged scientifically suitable for publication and will be formally accepted for publication once it meets all outstanding technical requirements.

Kind regards,

Wan-Long Chuang, M.D., Ph.D.

Academic Editor

PLOS ONE
---

## [Editor Report · Acceptance letter]

27 Oct 2020

PONE-D-20-17829R2 

Efficacy and safety of short-term therapy with indigo naturalis for ulcerative colitis: An investigator-initiated multicenter double-blind clinical trial 

Dear Dr. Suzuki:

I'm pleased to inform you that your manuscript has been deemed suitable for publication in PLOS ONE. Congratulations! Your manuscript is now with our production department. 

Kind regards, 

on behalf of

Dr. Wan-Long Chuang 

Academic Editor

PLOS ONE